# CX-5461 Preferentially Induces Top2α-Dependent DNA Breaks at Ribosomal DNA Loci

**DOI:** 10.3390/biomedicines12071514

**Published:** 2024-07-08

**Authors:** Donald P. Cameron, Jirawas Sornkom, Sameerh Alsahafi, Denis Drygin, Gretchen Poortinga, Grant A. McArthur, Nadine Hein, Ross Hannan, Konstantin I. Panov

**Affiliations:** 1ACRF Department of Cancer Biology and Therapeutics, Division of Genome Sciences and Cancer, The John Curtin School of Medical Research, The College of Health and Medicine, The Australian National University, Canberra, ACT 2601, Australia; donald.cameron@ki.se (D.P.C.); nadine.hein@anu.edu.au (N.H.); 2Division of Cancer Research, Peter MacCallum Cancer Centre, Melbourne, VIC 3000, Australia; jsornkom@its.jnj.com (J.S.); gretchen.poortinga@petermac.org (G.P.); 3Sir Peter MacCallum Department of Oncology, University of Melbourne, Melbourne, VIC 3000, Australia; grant.mcarthur@petermac.org; 4School of Biological Sciences, Queen’s University Belfast, Belfast BT9 5DL, UK; salsahafi01@qub.ac.uk; 5Pimera Therapeutics, 7875 Highland Village Place, Suite 412, San Diego, CA 92129, USA; denis.drygin@gmail.com; 6Department of Biochemistry and Molecular Biology, University of Melbourne, Parkville, VIC 3053, Australia; 7Department of Biochemistry and Molecular Biology, Monash University, Clayton, VIC 3800, Australia; 8School of Biomedical Sciences, University of Queensland, St Lucia, QLD 4072, Australia; 9Patrick G Johnston Centre for Cancer Research, Queen’s University Belfast, Belfast BT9 7AE, UK

**Keywords:** ribosome biogenesis, topoisomerase, DNA damage pathway, double-strand breaks, RNA Polymerase I, nucleolar surveillance pathway, p53, CX-5461

## Abstract

While genotoxic chemotherapeutic agents are among the most effective tools to combat cancer, they are often associated with severe adverse effects caused by indiscriminate DNA damage in non-tumor tissue as well as increased risk of secondary carcinogenesis. This study builds on our previous work demonstrating that the RNA Polymerase I (Pol I) transcription inhibitor CX-5461 elicits a non-canonical DNA damage response and our discovery of a critical role for Topoisomerase 2α (Top2α) in the initiation of Pol I-dependent transcription. Here, we identify Top2α as a mediator of CX-5461 response in the murine Eµ-*Myc* B lymphoma model whereby sensitivity to CX-5461 is dependent on cellular Top2α expression/activity. Most strikingly, and in contrast to canonical Top2α poisons, we found that the Top2α-dependent DNA damage induced by CX-5461 is preferentially localized at the ribosomal DNA (rDNA) promoter region, thereby highlighting CX-5461 as a loci-specific DNA damaging agent. This mechanism underpins the efficacy of CX-5461 against certain types of cancer and can be used to develop effective non-genotoxic anticancer drugs.

## 1. Introduction

Transcription of ribosomal DNA (rDNA) by RNA Polymerase I (Pol I), a critical step of ribosome biogenesis (RiBi), occurs in the nucleolus, a membraneless subnuclear compartment. Nucleoli form around actively transcribed ribosomal RNA (rRNA) genes, and changes in their size, number, and morphology have been linked to carcinogenesis for more than 100 years [1]. Correspondingly, hyperactivation of Pol I transcription has been documented in multiple cancer types, where it was associated with poor prognosis and intrinsic drug resistance [2,3]. Despite certain reservations due to the fundamental role that Pol I transcription plays in cellular biology, the idea of therapeutically targeting Pol I transcription, in particular, for cancer treatment, has gained much attention over the last decade (see for reviews [2,4,5]). As a result, small molecules that selectively inhibit rRNA transcription were developed (see for the latest review [2]) and the therapeutic potentials of this strategy were convincingly demonstrated [6,7,8,9,10,11]. Furthermore, the discovery that many common anticancer chemotherapeutic agents can dysregulate RiBi further strengthened their feasibility to therapeutically target what was once considered an undruggable “housekeeping” process that is essential to all cells [12,13]. CX-5461 was the first small-molecule selective inhibitor of Pol I transcription initiation [5] that progressed from pre-clinical development and testing to clinical trials [14,15], and recently received FDA designation to initiate clinical studies in patients with advanced solid tumors that carry BRCA1/2, PALB2 and other homologous recombination gene mutations. We and others demonstrated that inhibition of Pol I transcription by CX-5461 activates the nucleolar surveillance pathway (NSP) and upregulates stress signaling cascades that induce cell cycle arrest, cell differentiation, senescence, and/or cell death in tumor cells both in vitro and in vivo [6,8,16,17]. Intriguingly, CX-5461 is also able to elicit a DNA damage response (DDR), which is typically associated with DNA damaging agents (DDA) such as Topoisomerase II (Top2)poisons Etoposide and Doxorubicin [16,17]. However, the exact mechanism underlying DDR activation has not been identified. Recently, a number of studies proposing alternative mechanisms of action (MOAs) for CX-5461 [18,19,20,21,22,23] were published, but some results of these studies were, in part, contradictory (for the latest review see [2] and this manuscript). Two recent publications implicated Top2α poisoning as the MOA for CX-5461 anticancer activity [21,23]. Although here we confirmed that CX-5461 inhibits Top2α in a cell-free assay, we did not observe genome-wide damage as a result of the early response to CX-5461 treatment. In contrast to other Top2α poisons, CX-5461-elicited DNA damage is initially limited to the region adjacent to the rDNA promoter and we provide evidence that this targeted DNA damage is mediated by Pol I-associated Top2α. This suggests that CX-5461-induced DNA damage is associated with active rDNA repeats that prevail in cancer cells [24], and that this may be one of the reasons behind the anticancer selectivity of the drug.

## 2. Materials and Methods

### 2.1. Cell-Culture Conditions

Eμ-*Myc* GFP cells were cultured in vitro in Anne Kelso complete media (high glucose Dulbecco’s modified Eagle’s media (DMEM; Gibco-BRL, Gaithersburg, MD, USA)) supplemented with 10% heat inactivated fetal bovine serum (FBS; Sigma-Aldrich, Burlington, MA, USA), 1× GlutaMAX (Gibco-BRL, Gaithersburg, MD, USA)), 1× Antibiotic-Antimycotic (Gibco-BRL, Gaithersburg, MD, USA)), 1 mM L-asparagine (Sigma-Aldrich, Burlington, MA, USA), and 50 μM β-mercaptoethanol (Sigma-Aldrich, Burlington, MA, USA) at 37 °C in 10% CO_2_. Cells were plated in 6-well Falcon polystyrene plates (Thermo Fisher Scientific, Waltham, MA, USA) at 100,000–500,000 cells/well in 5 mL of media and passaged every two to three days. HTETOP cells (derivative of human fibrosarcoma cell line HT1080 [25]) were provided by Andrew Porter (Imperial College, London) and cultured in DMEM high glucose (4.5 g/L) with non-essential amino acids (Gibco-BRL, Gaithersburg, MD, USA)) supplemented with 10% FBS (Sigma-Aldrich, Burlington, MA, USA), 1 mM sodium pyruvate (Gibco-BRL, Gaithersburg, MD, USA)), 1× L-glutamine (Gibco-BRL, Gaithersburg, MD, USA)).

### 2.2. In Vivo Experiments

All animal work was conducted after approval by the Animal Ethics Committees at the Peter MacCallum Cancer Centre project E462 and the Australian National University projects A2012/15 and A2012/13 had been received. In vivo transplantation of Eμ-*Myc* lymphoma cells into recipient mice was performed as described previously [26]. Briefly, cells were defrosted, washed once, and resuspended in sterile phosphate-buffered solution (PBS) at a concentration of 1 × 10^6^ cells/mL. Then, 200,000 cells (200 μL) were injected into the tail vein of each mouse. To determine tumor engraftment, peripheral blood was extracted by either retro-orbital or tail-vein bleeding, and red blood cells were lysed by 5 min incubation at 37 °C in red blood cell lysis buffer (144 mM NH_4_Cl, 17 mM Tris pH 7.65). The tumor burden was monitored as the percentage of circulating GFP-expressing lymphoma cells in the peripheral blood, and was evaluated by flow cytometry. Animals were randomized into treatment groups when the average circulating tumor burden per cohort exceeded 10% of WBS.

CX-5461 (Synkinase, San Diego, CA, USA) was dissolved in 50 mM NaH_2_PO_4_, pH 4.5 and delivered by oral gavage. Mice were treated every three days with 35 mg/kg, or with 30 mg/kg CX-5461 in combination with everolimus. When in vivo shRNA induction was required, doxycycline was added to the food (600 mg/kg) and water (2 mg/mL + 2% sucrose). When mice reached a defined ethical endpoint, the animals were euthanized by cervical dislocation. Tumor tissue was harvested by crushing the axillary, brachial, and inguinal lymph nodes and washing the cells in sterile PBS + 2% FBS. For sequencing, tumor cells were further enriched by laying the tumor cell suspension over Ficoll-Paque (GE Healthcare, Hatfield, UK), spinning at 800 g for 20 min without centrifuge brakes, and harvesting the buffy coat. DNA and RNA were extracted using the DNeasy Blood and Tissue Kit (Qiagen, Hilden, Germany) and the RNeasy Mini Kit (Qiagen, Hilden, Germany), respectively.

### 2.3. Cloning and shRNA Transfection

To knockdown Top2α expression in Eμ-*Myc* lymphoma cells, we used a modified version of the RT3-REVIR construct (RT3-REBIR). Cell transduced RT3-REBIR constitutively expresses BFP (instead of the original mVenus) allowing for FACS selection. Upon induction by doxycycline, transduced cells expressed the dsRed fluorescent protein and the short hairpin RNA (shRNA), allowing the identification of cells expressing the hairpin construct [27,28]. All shRNA sequences used are listed in Appendix A. The shRNA vectors were packaged using the Phoenix E (PE) ecotropic retrovirus producer cells (ATCC) as described earlier [29].

Retrovirus-containing supernatant was harvested from the PE cells 24 and 48 h after media change, and the virus was concentrated to 1 mL/plate using Amicon Ultra-15 centrifugal filters (Merck Life Science UK Limited, Gillingham, UK). A 6-well plate was coated with 1 mL/well of a 30 μg/mL solution of recombinant human fibronectin (RetroNectin, TaKaRa, Kusatsu, Japan) in sterile PBS for 2 h at 4 °C. After washing the wells with 2% bovine serum albumin (BSA) in sterile PBS, 1 mL of concentrated virus was centrifuged onto the wells at 2000× *g* for 1 h at room temperature (RT). Then, 500,000 Eμ-*Myc* cells were added to each well in 2 mL of media and the plates were centrifuged at 400× *g* for 1 h at RT. This process was repeated after 24 h using the second viral supernatant from the PE cells. After passaging the transduced Eμ-*Myc* cells twice, cells with high BFP expression were isolated by FACS and expanded for tumor banking.

### 2.4. Quantitative Reverse Transcription PCR

Cells were harvested on ice and centrifuged at 400× *g* for 5 min at 4 °C. After washing with ice-cold PBS, RNA was extracted from 0.5–5 × 10^6^ cells using the NucleoSpin RNA kit (Macherey-Nagel, Dueren, Germany). The RNA concentration was determined by spectrophotometry using the NanoDrop 2000 (Thermo Fisher Scientific, Waltham, MA, USA). To synthesize complementary DNA (cDNA), 1–5 μg of RNA was incubated with 10 nmol dNTPs (Promega, Madison, WI, USA) and 50 pmol Random Primers (Promega, Madison, WI, USA) at 65 °C for 5 min, then placed on ice for 1 min. This solution was made up to 20 μL in 1× First Strand buffer (Thermo Fisher Scientific, Waltham, MA, USA) with 10 mM DTT (Promega, Madison, WI, USA), 40U RNasin ribonuclease inhibitor (Promega, Madison, WI, USA) and 60U Superscript III Reverse Transcriptase (Thermo Fisher Scientific, Waltham, MA, USA). The solution was incubated at RT for 5 min, then 50 °C for 1 h, then 70 °C for 15 min. The cDNA solution was diluted to 500 μL in RNase-free H_2_O before real-time quantitative PCR (qPCR). RNA abundance was determined using the 96-well StepOnePlus system (Applied Biosystems, Waltham, MA, USA). In each well, 8 μL of cDNA was combined with 2 μL of 1 mM stock of the forward and reverse primers and 10 μL of the Fast SYBR Green Master Mix (Thermo Fisher Scientific, Waltham, MA, USA). The samples were amplified by incubation at 95 °C for 10 s followed by 60 °C for 30 s, repeated for 40 cycles. Expression of each target was normalized to beta-2-microglobulin as expression of this gene remained consistent independent of drug treatment. The primers used are listed in Appendix A.

### 2.5. Western Blot Assay

After harvesting and washing twice by PBS, 1–5 × 10^6^ cells were resuspended in Western Solubilization Buffer (WSB; 20 mM HEPES pH 7.9, 500 μM EDTA, 2% SDS) and incubated at 95 °C for 5 min. Cells were then sheared by passing lysates ten times through a 26G needle. Protein concentrations were determined using the DC Protein Assay kit (Bio-Rad, South Granville, Australia). Protein gels were self-made with the exception of NuPAGE 3–8% Tris-acetate gels (Thermo Fisher Scientific, Waltham, MA, USA) used to resolve p-ATM. The resolving portion of the self-made gels consisted of 6–10% AA. After gel electrophoresis, proteins were transferred to a 0.45 μm PVDF membrane using the Trans-Blot Turbo RTA Mini LF PVDF Transfer Kit (Bio-Rad, South Granville, Australia). After incubating the membrane in Western blot blocking buffer (5% skim milk powder in Tris-buffered saline (TBS) with 0.1% Tween-20) for 1 h at RT, the membrane was placed in a 50 mL Falcon tube (Thermo Fisher Scientific, Waltham, MA, USA) containing 3 mL of blocking buffer with the primary antibody, then incubated on a blot roller overnight at 4 °C. After washing the membrane three times for 5 min in TBS with 0.1% Tween-20 (TBST), the membrane was incubated with the secondary antibody for 1 h at RT. Then, the membrane was washed again before the antibody-bound proteins were detected by chemiluminescence using the Clarity Western ECL Blotting Substrate (Bio-Rad) and imaged with the Gel Doc XR+ System (Bio-Rad, South Granville, Australia). In all cases, equal protein loading was confirmed by probing for housekeeping proteins β-actin or tubulin. Protein abundance was determined by quantitating band intensity using Image Lab v5.1 (Bio-Rad, South Granville, Australia). All antibodies used are listed in Appendix A.

### 2.6. Dose–Response Assay

Eμ-*Myc* cells were plated at 10,000 cells/well in the central 60 wells of a 96-well plate (Greiner). The following day, cells were treated in triplicate with either media, drug vehicle or varying doses of drug ranging from 50 pM to 30 μM in a final volume of 200 μL. Drugs assayed included CX-5461 (Synkinase, San Diego, CA, USA), Actinomycin D (Sigma-Aldrich, Burlington, MA, USA), BMH-21 (Medkoo, Durham, NC, USA), Etoposide (Sigma-Aldrich, Burlington, MA, USA), Doxorubicin (Sigma-Aldrich, Burlington, MA, USA), Camptothecin (Sigma-Aldrich, Burlington, MA, USA), KU-55993 (SynMedChem, Sichuan, China) and TMPyP4 (Santa Cruz Biotechnology, Dallas, TX, USA). After 24 h, either the number of live cells/well was measured by FACS or cell viability was determined using alamarBlue reagent (Thermo Fisher Scientific, Waltham, MA, USA). The live cell number was determined by resuspending each well in 200 μL PBS with 2% FBS and 10 μg/mL propidium iodide (PI), then performing volumetric FACS using the FACSVerse (BD Biosciences) to count the number of PI-negative cells. Cell viability was determined by adding 20 µL of alamarBlue reagent (Thermo Fisher Scientific, Waltham, MA, USA) to each well, followed by 3 h incubation and measuring absorbance at 570 nm wavelength normalized to absorbance at 600 nm. The average values for each dose group were plotted as log-transformed drug dose on the *x*-axis versus normalized live cell number/cell viability on the *y*-axis, where the readout from the vehicle-treated cells was set to 100%, and the readout with the lowest value was set as 0%. The line of best fit was then plotted using the “log(inhibitor) vs. normalized response—Variable slope” model in Prism 7 (GraphPad Software, Boston, MA, USA). The half-maximal growth inhibitory concentration (GI_50_) was calculated as the mid-point of the line of best fit where the drug has inhibited cell number/viability by 50% with respect to vehicle-treated cells.

HTETOP cells were plated in the central 60 wells of a 96-well plate (Greiner Bio-One, Stonehouse, UK) at density of 3 × 10^3^ cells/well. Following overnight incubation, cells were treated in triplicate with either the media, a drug vehicle, or varying doses of the drug in a final volume of 110 µL. The drugs assayed included CX-5461 (Synkinase, San Diego, CA, USA), Actinomycin D (Sigma-Aldrich, Burlington, MA, USA), BMH-21 (Medkoo, Durham, NC, USA), Etoposide (Sigma-Aldrich, Burlington, MA, USA). After a defined growth period (24, 48 or 72 h), cell viability was determined by incubating the plate for a further 30 min after the addition of 10 μL Presto Blue reagent (Thermo Fisher Scientific, Waltham, MA, USA) to each well, then measuring fluorescence (excitation: 560 nm; emission: 590 nm). The average values for each dose group were plotted by log-transformed drug dose on the *x*-axis and normalized live cell number/cell viability on the *y*-axis, where 100% corresponds to the readout from the vehicle-treated cells and 0% corresponds to the readout from growth media. The line of best fit was then plotted using the “log(inhibitor) vs. response—Variable slope” model in Prism 9 (GraphPad Software, Boston, MA, USA). The half-maximal inhibitory concentration (IC_50_) was calculated as the mid-point of the line of best fit where the drug has inhibited cell number/viability by 50% with respect to vehicle-treated cells.

### 2.7. Immunofluorescence Imaging

Cells at a concentration of 1 × 10^6^ were fixed in 5 mL 2% paraformaldehyde (Pierce, Waltham, MA, USA) in PBS for 5 min at RT, then washed twice and resuspended in 500 μL PBS. Fixed cells at a concentration of 3 × 10^5^ (150 μL) were then cytospun onto Superfrost Plus microscopy slides (Thermo Fisher Scientific, Waltham, MA, USA) using cytology funnels in yielding 6 mm spots. The slides were then air-dried and stored at −20 °C. All incubations were undertaken with a volume of 200 μL/slide in a humidified chamber to prevent evaporation, and all washes were performed three times for 5 min each. Cells were permeabilized by incubation in PBS with 0.3% Triton X-100 (PBST) for 10 min. If stained for G4 DNA, the cells were incubated with 40 μg/mL RNase A in PBS for 1 h at 37 °C, then washed in PBS before proceeding with standard IF staining as described below. Cells were incubated with IF blocking solution (5% goat serum in PBST) for 30 min at RT, then with the primary antibody diluted in IF blocking solution for 1 h at 37 °C. After washing with PBS, cells were incubated with secondary antibodies conjugated to AlexaFluor 488/594 fluorophores in IF blocking solution for 1 h at 37 °C. Cells were washed in PBS, then incubated for 5 min with 2 μg/mL 4′,6-diamidino-2-phenylindole (DAPI) in PBS at RT. Once the slides were washed, they were mounted in VectaShield Antifade Mounting Medium (Vector Laboratories, Newark, CA, USA) under coverslips and sealed with nail varnish. All antibodies used are listed in Appendix A.

Cells were imaged either in 2D using the Olympus BX61 or in 3D using the Zeiss Confocal LSM 780. Quantitation of nuclear staining was made by using DAPI signal as a nuclear mask, then measuring the total pixel intensity of the target within a nucleus using CellProfiler [30]. The ImageJ DiAna plugin was used to measure distances between γH2AX and fibrillarin foci in 3D images [31].

### 2.8. Top2 Activity Assay

The technique for extraction of Top2 enzymes from Eμ-*Myc* was adapted from a published method [32]. Briefly, 8 × 10^6^ Eμ-*Myc* cells were collected and washed twice in ice-cold PBS. Nuclei were isolated by incubation in 10 mL NP-40 buffer (10 mM Tris pH 7.4, 10 mM MgCl_2_, 10 mM NaCl, 0.5% IGEPAL CA-630 (Sigma-Aldrich, Burlington, MA, USA), 1× cOmplete Mini Protease Inhibitor Cocktail (Sigma-Aldrich, Burlington, MA, USA)) for 10 min on ice. Nuclei were centrifuged at 500× *g* for 10 min at 4 °C, washed once in 1 mL ice-cold PBS, then incubated in 200 μL high-salt extraction buffer (20 mM Tris pH 7.5, 1 mM MgCl2, 350 mM KCl, 10% glycerol, 1 mM DTT, 1× cOmplete Mini Protease Inhibitor Cocktail) overnight at 4 °C. Nuclear debris was pelleted at 15,000 g for 10 min and the supernatant was serially diluted for the decatenation assay. When the Top2 inhibitory activity of drugs was tested, recombinant Top2α enzyme (TopoGen, Buena Vista, CO, USA) was used in place of cell-extracted Top2. Top2 activity was determined using the Topoisomerase II Assay Kit (TopoGen, Buena Vista, CO, USA) to quantify the decatenation of kinetoplast DNA (kDNA) from Crithidia fasciculata by Type II topoisomerases. Dilutions of the nuclear protein extracts or a fixed amount of recombinant Top2α were incubated with 200 ng kDNA in the provided buffer (with serial dilutions of candidate Top2 inhibitors if required) for 30 min at 37 °C. Then, the reaction was stopped using the stop buffer provided in the kit and the product was run on a 1% agarose gel for 45 min. Top2 activity was calculated as the intensity of the decatenated DNA bands able to escape the well as a percentage of the total DNA signal from the gel lane.

### 2.9. TARDIS Assay

This assay was adapted from previously published methods [33,34]. Microscopy slides were incubated in 1M HCl overnight at 55 °C, then washed in ultrapure water and stored until use in 100% ethanol to ensure the adherence of agarose to the slides. A 200 μL layer of molten 60 °C 0.5% normal-melting-temperature agarose (NMA) was pipetted onto the slide and allowed to cool and set on ice under a 22 × 40 mm coverslip. Then, 1 × 10^6^ Eμ-*Myc* cells were harvested, washed, and resuspended in 30 μL ice-cold PBS.

The cell suspension was heated at 37 °C for 30 s before combining with 30 μL of molten 37 °C 2% low-melting-temperature SeaPrep agarose (Lonza, Sydney, Australia). This agarose–cell mixture 47 was rapidly pipetted on top of the NMA layer and set under a coverslip on ice for 30 min. Once set, the coverslip was carefully removed and the slide was incubated in lysis buffer (1% SDS, 80 mM NaH_2_PO_4_ pH 6.5, 10 mM EDTA, 1 mM DTT, 1× cOmplete Mini Protease Inhibitor Cocktail) for 30 min at RT. Slides were then incubated in high salt buffer (1M NaCl, 1× cOmplete Mini Protease Inhibitor Cocktail) for 30 min at RT to remove all cellular components apart from the DNA and covalently attached macromolecules.

After washing 3 times for 5 min in PBS, the slide was incubated in TARDIS blocking buffer (1% BSA, 0.1% Tween-20 in PBS) with a 1:1000 dilution of Top2α primary antibody for 1 h at RT. Slides were washed as before, then incubated in TARDIS blocking buffer supplemented with a 1:1000 dilution of fluorophore-tagged secondary antibody for 1 h at RT. The slides were washed again before staining DNA with DAPI and mounting. Slides were imaged, analyzed, and quantitated as described above.

### 2.10. DNA Double-Strand Break Labelling

This assay was adapted from previously published methods [35,36]. Briefly, HTETOP cells (two 15 cm dish at 70% confluency were used per single experimental point) were washed with 37 °C PBS and fixed in 10 mL per dish of the non-DNA-damaging Streck Fixative (0.15 M 2-bromo-2-nitropropane-1,3-diol, 0.1 M diazolidinyl urea, 0.04 M zinc sulfate heptahydrate, 0.01 M sodium citrate dihydrate and 50 mM EDTA) for 20 min on rotating platform at RT. Next, cells were permeabilized by further incubation on the rotating platform in 10 mL of PM buffer (0.1 M Tris-HCl, pH 7.5; 50 mM EDTA; 1% Triton X100) for 15 min at RT.

Cells were carefully harvested with a large cell scraper in 10 mL PBS and pelleted by centrifugation at 220× *g* for 5 min and resuspended in 2 mL of 1× TdT buffer (50 mM potassium acetate, 20 mM Tris-acetate, 10 mM magnesium acetate, 250 μM cobalt (II) chloride, pH 7.9). Cells were washed three times with 2 mL of the same buffer. Cells were subsequently incubated in 1 mL reaction buffer (1× TdT buffer, 0.005% Triton X-100, 100 μM 5-Bromo-2′-Deoxyuridine 5′-Triphosphate (BrdUTP, Sigma-Aldrich, Burlington, MA, USA), 500 µM Deoxyadenosine 5′-Triphosphate (dATP, Sigma-Aldrich, Burlington, MA, USA), 500U terminal deoxynucleotidetransferase (Promega)) for 90 min at 37 °C on a rotating wheel. Cells were washed twice with 2 mL ice-cold PBS and lysed in 300 μL Lysis buffer (50 mM Tris.HCl pH 8.0, 125 mM NaCl, 10 mM EDTA, 1% Triton X-100, 0.1% sodium deoxicholate, 0.5% sodium N-lauroylsarcosine) for 15 min at 37 °C yielding partially purified nucleoli. Nucleoli were then precipitated by centrifugation at 2000× *g* for 5 min 4 °C and resuspended in 250 μL of the same Lysis buffer.

Chromatin was sheared using the Bioruptor (Diagenode, Seraing (Ougrée), Belgium) that was set to High Power, 30 sec ON, 30 sec OFF. Sonication time (5 min) was determined experimentally (by electrophoresis of sheared DNA in 2% agarose gel) to yield 500 bp of average size fragments. Shearing was performed at the temperature not exceeding 10 °C. Sheared chromatin was incubated overnight with anti-BrdU antibodies (10 µg per experimental point, BD Biosciences, Wokingham, UK). Labelled DNA fragments were captured with 60 μL of Dynabeads Protein A magnetic beads (Thermo Fisher Scientific, Waltham, MA, USA) for 4 h at room temperature. Beads were washed five times in 450 μL of RIPA ChIP buffer [37] and twice in 200 μL TE buffer using a Precipitor (Abnova, Taipei City, Taiwan). DNA elution was performed as described previously [38].

Purified DNA was analyzed by qPCR using QuantiFast Multiplex PCR Mix (Qiagen) and the rDNA promoter-specific combination of primers and probe (see Appendix A) were analyzed using LC480-II cycler (Roche, Burgess Hill, UK). All PCR reactions were performed in triplicate, and standard deviation was calculated from three independent experiments.

### 2.11. Metabolic Labelling

Cells at a concentration of 5 × 10^5^ were treated with 500 μCi ^32^P orthophosphate for 20 min prior to harvest. RNA was extracted using the NucleoSpin RNA kit (Macherey-Nagel, Dueren, Germany), then 5 μg was loaded onto a 1.2% agarose MOPS/formaldehyde gel with ethidium bromide and run for 16 h at 50 V. The 28S band was visualized by UV excitation, then the gel was dried and exposed to a phosphoimager screen overnight and developed using the Typhoon TRI variable mode imager (GE Healthcare, Hatfield, UK) to visualize the 47S pre-rRNA band.

## 3. Results

### 3.1. Topoisomerase 2α Mediates Resistance of Cancer Cells to CX-5461

#### 3.1.1. Mutations in Top2α Mediate Resistance to CX-5461 in Eμ-Myc Lymphoma

We have previously demonstrated that Eμ-*Myc* lymphoma-bearing mice on CX-5461 therapy exhibit significantly increased survival, but eventually relapse with tumors that are intrinsically resistant to subsequent CX-5461 challenge [6,26]. To investigate the mechanism of resistance to CX-5461 in Eµ-*Myc* lymphoma, we transplanted previously characterized p19ARF−/−; Bcormut; Eµ-*Myc* clone #4242 [39] into C57/BL6 mice, then treated animals with established disease with either CX-5461 or a drug-delivery vehicle until the mice succumbed to disease. Tumor DNA was extracted from the lymph nodes of three CX-5461-treated animals that had relapsed during therapy and three vehicle-treated mice. Whole exome sequencing was performed to identify mutations that may have conferred resistance to CX-5461 treatment. Strikingly, the only recurrent mutated gene across all three tumors was that encoding Top2α, which exhibited seven independent mutations across the three tumors (Figure 1A and Appendix A). Four of the mutations were predicted to encode a truncated protein product, while the other three missense mutations were in the key catalytic TOPRIM and WHD domains. Therefore, all mutations were predicted to decrease Top2α cellular activity.

For further characterization, we used an Eµ-*Myc* clone derived from one of the CX-5461-resistant tumors containing a heterozygous Top2α mutation at base pair c.3921-3922, where a T insertion leads to a reading frame shift at the K1266 residue (T2A^+/K1266^). Clone T2A^+/K1266^ exhibited a reduction in Top2α expression (Figure 1B,C), which was associated with a loss of cellular Top2α activity as measured by decatenation assay (Figure 1D). In vitro, the T2A^+/K1266^ clone was roughly 2.5-fold more resistant to CX-5461 than a drug-naïve Top2α wild-type (T2A^WT^) Eµ-*Myc* clone (Figure 1E). Furthermore, in vivo CX-5461 treatment of mice with established T2A^+/K1266^ lymphoma did not significantly increased the median survival (compare Figure 1F,G).

#### 3.1.2. Top2α Mediates Resistance to CX-5461 in Different Cellular Models

To verify the link between the level of Top2α expression and sensitivity to CX-5461 treatment, T2A^WT^ lymphoma cells were transduced with a modified version of RT3-REVIR doxycycline-inducible knockdown system targeting Top2α (shT2A), alongside a non-targeting (shNT) control [27,28]. Doxycycline treatment for 48 h induced a ~60% reduction in Top2α expression at both the mRNA and protein level in the shT2A cells relative to the shNT cells, thus phenocopying the decrease in Top2α expression exhibited by the T2A^+/K1266^ cells (Appendix A).

Critically, a decrease in the level of Top2α protein led to a concomitant increase in the resistance of shT2A cells to CX-5461 treatment compared to the control in both cells and animal models (Appendix A). Interestingly, while T2A^+/K1266^ cells were less sensitive to Top2α poisons, namely Doxorubicin and Etoposide, than T2A^WT^ cells, there was no significant difference in sensitivity of these cells to Pol I inhibitor BMH-21 (Appendix A).

To test whether the link between Top2α expression level and CX-5461 resistance is cell type (cancer type)-specific, we measured a CX-5461 dose–response using the HTETOP cell line, a modified human fibrosarcoma HT1080 cell line where Top2α expression can be >99.5% ablated by treatment with doxycycline (Dox) [25] (Appendix A). Similar to shT2A and shNT cells, we observed a significant difference in sensitivity to CX-5461 between cells lacking and expressing Top2α (Appendix A); there was no detectable effect of Top2α depletion on efficacy of BMH-21 (Appendix A), but a striking 10-fold decrease in sensitivity to Etoposide (Appendix A). These results suggest that the role of Top2α in mediating CX-5461 cytotoxicity is cell-type-independent.

### 3.2. Alterations in Top2α Level/Activity Does Not Affect the Ability of CX-5461 to Inhibit rRNA Synthesis, but Modulates the Level of CX-5461-Induced DDR

#### 3.2.1. The Expression Level of Top2α Does Not Affect the Pol I Inhibitory Activity of CX-5461

CX-5461 has been previously demonstrated to inhibit Pol I transcription by displacing the transcription factor SL1 from the rDNA promoter at concentrations close to the GI_50_ [5]. To investigate whether CX-5461 Pol I inhibitory activity could be modulated by the level of Top2α protein, we measured the effect of CX-5461 on the rate of rRNA synthesis in T2A^WT^ and T2A^+/K1266^ lymphoma cells using metabolic labelling as described previously [6]. The results showed no significant differences in the rate of rRNA transcription following treatment between the two cell lines (Figure 2A, compare top and bottom panels and Appendix A).

The depletion of Top2α also does not significantly affect the inhibition of rRNA synthesis by CX-5461 in HTETOP cells (see Appendix A), strongly suggesting that Top2α does not directly modulate the capacity of CX-5461 to inhibit Pol I transcription. Importantly, the classical Top2α poisons such as Etoposide or Top2α depletion also have no effect on the level of the steady-state rRNA synthesis as we demonstrated earlier [35].

#### 3.2.2. The Expression Level of Top2α Modulates the Level of DDR Induced by CX-5461 Treatment

We, and subsequently others, have demonstrated that CX-5461 treatment induces a DNA damage response (DDR) leading to S-phase delay and G2 cell cycle arrest in the absence of genome-wide DNA damage [16,17,41]. Therefore, we investigated whether the reduction in Top2α affects CX-5461 ability to activate the DDR in both the T2A^WT^ and T2A^+/K1266^ cell lines.

We found that reduced Top2α protein level led to a marked reduction in ATM activation (Ser1981 phosphorylation [42,43]), suggesting a role for Top2α in DDR pathway activation by CX-5461 (Figure 2B). Surprisingly, CX-5461 induces the p53-target genes p21 and Puma in a Top2α-dependent manner, regardless of the relatively modest effect on p53 stabilization and its phosphorylation at Ser15, a residue that is downstream of ATM [44,45] (Figure 2B–D and Appendix A). This finding suggests that the downstream Top2α-dependent response can be either mediated independently of p53 or through activation of p53 by phosphorylation at alternative sites.

To distinguish the ATM-dependent DDR downstream effects of CX-5461 that were described earlier [8] from other signaling pathways, we pre-treated cells with KU-55933 (ATMi), a specific inhibitor of ATM [46], before treating with CX-5461 for 3 h. Pre-treatment with ATMi reduces autophosphorylation of ATM at serine 1981 in response to CX-5461 treatment, confirming that ATMi inhibits ATM activation (Figure 2B and Appendix A). Furthermore, ATMi pre-treatment also significantly reduced CX-5461-driven p21 and PUMA upregulation (Figure 2C,D), while also moderately reducing the induction and phosphorylation of p53 (Figure 2B and Appendix A). Finally, ATMi pre-treatment resulted in the attenuation of the CX-5461 response in both cell lines, suggesting that ATM-dependent signaling potentiates the tumor-killing mechanism of CX-5461 irrespective of the Top2α status (Figure 2E).

Thus, our results strongly suggest that ATM is required to propagate the Top2α-dependent downstream apoptotic response to CX-5461.

### 3.3. Physiologically Relevant Concentrations of CX-5461 Do Not Cause Genome-Wide DNA Damage as an Early Response

#### 3.3.1. CX-5461 Is a Top2α Inhibitor That Does Not Induce Genome-Wide Top2α-Mediated DNA Damage

Top2α poisoning was recently suggested as the MOA for CX-5461’s anticancer activity [21,23]. Here, we confirmed that CX-5461 inhibits Top2α in vitro and in cells and tested whether CX-5461 induces DNA damage at the physiologically relevant concentrations in comparison to known Top2α poisons (e.g., Doxorubicin).

Indeed, we observed that CX-5461 inhibits decatenation of kinetoplast DNA (kDNA) to a similar degree as Doxorubicin demonstrating that CX-5461 acts as Top2α inhibitor (Appendix A). Top2 poisons such as Etoposide and Doxorubicin prevent the Top2α enzyme from re-ligating DNA after the initial double-strand break is made. This creates Top2α cleavage complexes (Top2cc) which, if not resolved, trigger a DDR, leading to apoptotic cell death and/or cell cycle arrest [47,48,49,50].

To assess whether CX-5461 treatment of Eµ-*Myc* cells causes Top2α-dependent DNA damage, we performed a TARDIS assay [33,34] (Figure 3A) that allows for the quantification of Top2cc foci after drug treatment. To inhibit the proteasomal degradation of Top2cc, T2A^WT^ cells were pre-treated with MG132 for 30 min [51,52] and then treated with the respective GI_90_ doses of CX-5461, Doxorubicin, or vehicle for 2 h before processing (Figure 3B). Critically, we used relatively high, but still physiologically relevant, concentrations of drugs that inhibited 90% of cell growth (30 nM for both drugs).

In cells treated with MG132 alone, only very few Top2cc foci formed, likely representing basal Top2α activity. In contrast, Doxorubicin treatment induced robust Top2cc foci formation, in line with the expected effects of Top2α poisons [53] (Figure 3B,C).

Strikingly, CX-5461 treatment had a negligible impact on cellular Top2cc foci levels when compared to the vehicle-treated cells during the course of experiment (Figure 3D). These data suggest that, contrary to predictions based on the decatenation assay, CX-5461 does not act as a classical Top2α poison, regardless of its ability to induce DDR [16,17].

Our earlier findings revealed the association of Top2α with the Pol I initiation complex and its involvement in pre-initiation complex (PIC) formation [35]. Building on this, we hypothesized that CX-5461 may initiate the DNA damage response (DDR) by preventing the re-ligation of DNA double-strand breaks induced by Pol I-associated Top2α by inhibiting its activity. At the physiologically relevant concentrations of the drug, this will result in a limited level of Top2cc formation at other genomic loci and may explain the correlation between Top2α-inactivating mutations and CX-5461 resistance.

To test this hypothesis, we aimed to identify the location of the DNA damage signal using antibodies specific to phosphorylated histone variant H2AX (termed γH2AX), a well-established marker for DNA damage foci [54,55]. In addition to γH2AX antibodies, we also used antibodies specific to fibrillarin to label rDNA loci [56,57] (Figure 3D). Remarkably, CX-5461-induced γ-H2AΧ foci were enriched 0.3 μm from the nucleolar periphery, while in Etoposide-treated cells, these foci were distributed across the nucleoplasm (Figure 3D–F).

These data imply that CX-5461 induces targeted DNA damage at nucleolar-adjacent regions, in contrast to Etoposide which elicits genome-wide DNA damage.

#### 3.3.2. CX-5461 Treatment Induces Targeted DNA Damage at the rDNA Loci

We observed that the Top2α-mediated γH2AX signal induced by CX-5461 treatment is predominantly localized to the nucleolar periphery (Figure 3D), suggesting that this DNA damage may be localized to rDNA.

To verify this, we labelled the double-strand DNA breaks (DSB) as described earlier [36]. In this method, the DNA ends labelled with modified nucleotides using a terminal transferase, which allows capturing of the labelled DNA for analysis.

For this experiment, we used the HTETOP cell line where Top2α expression can be >99.5% ablated by treatment with doxycycline (Dox) [25] (Appendix A). By treating the HTETOP cells with CX-5461, ActD, or Etoposide in the presence or absence of Dox, we determined whether drug-induced DNA DSBs are Top2α-dependent and where the DSBs are located within rDNA loci. To this end, the purified labelled DNA is analyzed by qPCR with primers and probes spanning eight regions across rDNA (Appendix A) as previously described [35]. The concentrations of CX-5461 and ActD were selected to effectively repress rRNA synthesis within 30 min after drug treatment (Appendix A). Importantly, CX-5461 concentration (2 µM) was similar to its GI_50_ value (1.48 µM) for HTETOP cells (Appendix A).

We found that CX-5461 treatment in cells expressing normal levels of Top2α led to a profound increase in DSB near the rDNA promoter region (Figure 4A, red bars). Notably, pre-treatment of HTETOP cells with Dox abrogated the induction of rDNA-specific DSBs by CX-5461, indicating that the observed phenomenon is Top2α-dependent (Figure 4A, dark red bars). Highly localized DSBs caused by CX-5461 contrast sharply with DSBs caused by the Top2α poison, Etoposide, where DSBs are randomly distributed along rDNA repeats (Figure 4B, red bars), and are also negated by Top2α depletion (Figure 4B, dark red bars) [35]. In contrast to CX-5461, ActD treatment did not alter DSB levels independently of Top2α expression status (Figure 4C, note scale difference between A, B, and D).

We also investigated the dynamics of DSB formation in cells treated with CX-5461 (Figure 4D) and discovered that DSBs occurred rapidly at the promoter region (within one hour after drug treatment) but were transient, disappearing within 4 h after treatment. Interestingly, that at the 4 h time point, DSBs become detectable in the non-transcribed IGS region. We propose that these observations probably reflect two independent processes: firstly, the depletion of the initiating form of Pol I associated with Top2α (resulting in DSB disappearance), and secondly, the CX-5461-driven inhibition of Top2α that is not associated with the Pol I transcription machinery (leading to the appearance of DSBs localized in the IGS).

In control experiments, we demonstrated that HTETOP treatment with Dox did not cause increase in the level of DSBs (Appendix A) and that the induction of DSBs by CX-5461 can be blocked by Top2α depletion at various time points (Appendix A). Moreover, ActD at concentrations that inhibit only Pol I transcription (3 nM), also did not cause an increase in the level of DSBs within 4 h of treatment (Appendix A).

Together, these data strongly suggest that the initial response to CX-5461 treatment (in addition to inhibition of Pol I transcription) is the induction of DNA damage specifically located at the rDNA promoter region, in contrast to the inhibition of Pol I transcription with ActD, which causes no DSBs formation as opposed to treatment with other Top2α poisons (Etoposide), which cause more widespread DNA damage.

### 3.4. Induction/Stabilization of G4 Structures in the Eμ-Myc Lymphoma Cellular Model Is Not Sufficient to Induce Cell Death In Vitro

In their 2017 manuscript, Xu and colleagues proposed a novel MOA for CX-5461, suggesting the stabilization of guanine quadruplex (G4) structures in telomeres by CX-5461 [18]. To investigate the link between the potential of CX-5461 to stabilize G4 structures intracellularly and its anti-tumor activity in Eμ-*Myc* murine B-cell lymphoma cells [6], we used tetra-*N*-methylpyridyl porphyrin (TMPyP4), a well-established G4 DNA stabilizer, as a positive control [58,59]. Importantly, in contrast to G4 ligands used by other groups (for instance pyridostatin, PDS), it has no Top2α inhibitory activity, allowing us to separate the effects of G4 binding/stabilization from Top2α inhibition [23,60].

Dose–response analysis demonstrated that TMPyP4 failed to have an impact on Eμ-*Myc* cell viability at concentrations up to 30 μM, while the data for CX-5461 were similar to our previously published results [61] (Appendix A).

Next, we performed G4 stabilization analysis using the same method and antibody (BG4) that was described in studies published by Xu et al. Eμ-*Myc* cells were treated for 1 h with either vehicle; 500 nM CX-5461 (a dose 50-fold higher than the observed GI_50_); or 10 µM TMPyP4 (a dose that had no impact on Eμ-*Myc* cells’ viability). The treatment was followed by immunofluorescent staining using an antibody specific to G4 DNA structures (Appendix A). As expected, TMPyP4 treatment resulted in a statistically significant increase in G4 stabilization in all treated cells, as demonstrated by an upward shift of the entire TMPyP4 box plot (Appendix A).

Notably, treatment of Eμ-*Myc* cells with a dose of CX-5461 markedly higher than the physiologically relevant concentration failed to enhance the stabilization of G4 structures (Appendix A). This indicates that, within the Eμ-*Myc* context, CX-5461 does not stabilize G4 DNA structures even at doses well beyond those inducing cell deaths. Furthermore, G4 stabilization by a ligand lacking Top2α inhibitory activity does not promote Eμ-*Myc* cell death.

## 4. Discussion

To analyze the underlying cause of acquired resistance to CX-5461 in Eμ-*Myc* lymphoma, we performed WGS on resistant tumors and identified Top2α as one of the most recurrently mutated genes. Although other genes such as NARF and COL6A6 (Appendix A) were also among frequently mutated genes, in this paper, we focused on Top2α. We demonstrated that resistance to CX-5461 was directly correlated with the level of Top2α protein and Top2 activity not only in mutated clones derived from a resistant tumor (Figure 1), but also in cell lines in which Top2α level was downregulated (Appendix A).

We demonstrated that resistance to CX-5461 was directly correlated with the level of Top2α protein and Top2 activity, not only in a mutated clones derived from a resistant tumor (Figure 1), but also in cell lines in which Top2α levels were downregulated (Appendix A).

We also definitively demonstrated that the ability of CX-5461 to inhibit Pol I transcription is not dependent on its ability to inhibit Top2a. For instance, cells lacking Top2α are 2–3-fold less sensitive to CX-5461 (Figure 1E, Appendix A) in cell viability assay, but this deficiency does not significantly affect CX-5461 Pol I inhibitory activity (Appendix A), suggesting that Top2α levels contribute to the growth inhibitory effect of CX-5461, but do not fully define it.

Importantly, at a concentration of CX-5461 equal to the GI_50_ concentration, more than 90% of Pol I transcription is inhibited within 1–2 h after drug treatment regardless of the Top2α level (compare the corresponding GI_50_ and IC_50_ values, Figure 1E, Appendix A). This suggests that the Top2α inhibitory activity of CX-5461 modulates its growth inhibitory activity in a Top2α-dependent manner, and 2–3-fold changes in drug sensitivity define the difference between CX-5461-resistant and CX-5461-sensitive tumors in a B-lymphoma model.

In summary, our results identify Top2α as (i) a major driver of resistance to CX-5461 in vivo (Appendix A and Figure 1), and (ii) a mediator of the anticancer activity of CX-5461 (Figure 1 and Appendix A). Furthermore, previous studies, including our own and others’, have indicated that the induction of DNA damage response (DDR) contributes to the therapeutic mechanism of action (MOA) of CX-5461 [16,17]. However, until now, the exact mechanism underlying CX-5461-induced DDR activation and its impact on overall cytotoxicity remained unclear. The identification of CX-5461 as a Top2α inhibitor, both in this study and elsewhere [21,23], provides a molecular mechanism for CX-5461-induced DDR activation and sheds light on the contribution of Top2α inhibitory activity of CX-5461 to its overall therapeutic activity.

In addition to this paper, two other groups have reported anti-Top2α activity of CX-5461, albeit the proposed mechanisms of anticancer activity differed [21,23]. Bruno and coworkers tested the cross-resistance between in vitro selected CX-5461-resistant Eµ-*Myc* clones and Doxorubicin as well as ActD. Conversely, the study by Bruni et al. revealed that CX-5461-resistant clones exhibited resistance to both Doxorubicin and CX-5461; the genetic alterations that conferred the resistance were not identified [21]. They also demonstrated direct inhibition of Top2α by CX-5461 in the decatenation assay, a finding consistent with our data (Appendix A).

Bruno and coworkers proposed that Top2α poisoning is the main mechanism of the antitumor activity of CX-5461 [21]. While their data are interesting, there are multiple lines of evidence indicating that as a therapeutic agent CX-5461 exhibits clinical properties quite different from known Top2α poisons. Both typical (e.g., Doxorubicin, Etoposide) or atypical (e.g., Vosaroxin) Top2α poisons share very similar clinical safety profiles, with the most commonly observed adverse events (AE) being leukopenia and nausea/vomiting [62,63,64]. In contrast, in clinical studies of CX-5461, the most commonly observed AEs were photosensitivity and palmar–plantar erythrodysesthesia [14,15]. Several other published studies highlight pharmacological differences between Top2α poisons and CX-5461. Unlike Top2α poisons whose activity correlates with p53 status [65,66], activity of CX-5461 was found to be p53-independent in the majority of analyzed tumor types and no p53-dependence was observed in the clinic [5,7,8,14]. Furthermore, CX-5461 demonstrated more potent activity in an aggressive MLL-ENL*Nras model of AML that was resistant to standard therapy, a regime of Cytarabine—Doxorubicin administered at MTD [8].

While both DNA damaging agents and Pol I transcription inhibitors can induce p53 stabilization, they manage it through different mechanisms. DNA damaging agents primarily rely on tumor suppressor Arf to sequester MDM2, leading to the stabilization of p53 [67], while treatment with Pol I inhibitors triggers nucleolar stress, which leads to the sequestration of MDM2 by ribosomal proteins (for review, see [68]). Zhang and co-workers generated mice carrying a single cysteine-to-phenylalanine substitution in the central zinc finger of Mdm2 (Mdm2C305F) that disrupts Mdm2′s binding to RPL11 and RPL5, leading to an attenuated p53 response and the perturbation of ribosome biogenesis [69].

In our recent publication [70], we used mouse embryonic fibroblasts (MEFs) from the Mdm2C305F mutant and wild-type mice to compare the induction of p53 in response to treatment with CX-5461, as well as controls. There was significantly lower induction of p53 in Mdm2C305F MEFs than in Mdm2WT MEFs by CX-5461, indicating that this compound activates p53 primarily through NSR. In contrast, Top2α poisons Doxorubicin and Etoposide-induced p53 in Mdm2C305F MEFs with a similar efficiency as in Mdm2WT MEFs, demonstrating that Top2α poisons activate p53 via the Mdm2-independent mechanism, again indicating a different MOA between CX-5461 and Top2α poisons (Figure 4D in [70]).

In line with these findings, our results indicate that the activity level of Top2α in both drug-naive and drug-resistant cells influences the activation level of DDR and, subsequently, the response to CX-5461 treatment. However, owing to the dual activity of CX-5461, cellular response, including the activation of 53 and its targets, is not entirely suppressed by low Top2α activity (see Figure 2B–E, Appendix A).

Therefore, based on these data, we conclude that while the anti-Top2α activity of CX-5461 is important, it is not the sole determinant of CX-5461 anticancer activity. Furthermore, the contribution of Top2α inhibitory activity to CX-5461’s overall anticancer efficacy is likely to vary across different types of malignancies. This view is reinforced by a recent study by Pan and colleagues, attributing the therapeutic activity of CX-5461 in high-risk neuroblastomas primarily to inhibition of Top2β [22]. It should be noted that Top2 poisons typically inhibit both isoforms (α and β), albeit with varying potency [50,71]. Notably, Top2β is the primary oncogenic Top2 isoform in *NMyc*-driven cancers such as neuroblastoma [72], and this may explain the distinct mechanisms of action of CX-5461 in neuroblastomas and in lymphomas.

Another potential CX-5461 MOA that is dependent on its Top2α inhibitory activity, was proposed by Bossaert and colleagues [23]. They linked CX-5461′s Top2α poisoning activity to its capability to bind/stabilize guanine quadruplex (G4) structures, a concept initially proposed by Xu and colleagues [18]. Bossaert et al. concluded that CX-5461’s Top2α poisoning activity results in the formation of transcription-dependent double-stranded breaks (DSB’s) associated with G4 structures within transcribed regions. However, this conclusion is not entirely consistent with the conclusions of Xu et al. They proposed that the increased sensitivity of cancer cells with defects in the HR pathway (e.g., loss of BRCA1/BRCA2) to CX-5461 results from CX-5461’s stabilization of G4 structures in telomeres, causing replication fork blocking and the induction of ssDNA gaps and breaks. Nonetheless, both groups postulate the important role of CX-5461’s G4 stabilizing activity in its mechanism of action.

In this study, we also examined whether CX-5461 induces G4 structures formation in the Eµ-*Myc* model and whether such induction triggers cell death. Our results (Appendix A) indicate that in the Eµ-*Myc* model, even at doses well above those that cause cell death, CX-5461 does not stabilize G4 structures (Appendix A). Moreover, induction of moderate level G4 formation, by the known G4 stabilizer TMPyP4 that lacks Top2α activity, does not promote cell death (Appendix A). Thus, at least in the Eu-*Myc* model of lymphoma, G4 stabilization is unlikely to mediate the therapeutic activity of CX-5461. This further highlights the potential differences in the mechanisms of action of CX-5461 across various malignancies.

It is well documented that rDNA integrity is strictly controlled by nucleolar surveillance mechanisms [73,74]. Previously, we have demonstrated that components of the DDR pathway associated with actively transcribed rDNA, which promotes a rapid and profound response to DNA damage [75]. Here, we examined the effects of physiologically relevant doses of CX-5461 on the induction of DNA damage and the formation of Top2α-DNA covalent complexes using TARDIS assay.

The results of this assay and the analysis of the subcellular localization of γ-H2AΧ foci in cells treated with CX-5461 and Top2α poisons (Figure 3) clearly showed that the number of covalent Top2α-DNA complexes in cells treated with a GI_90_ dose of CX-5461 was significantly lower than in cells treated with a GI_90_ dose of Top2α poison, etoposide (and Doxorubicin), after 2 h of drug exposure. Interestingly, γ-H2AΧ foci in CX-5461-treated cells were primarily associated with the nucleolar periphery compared with etoposide-induced foci (Figure 3F). It is a well-known phenomenon that rDNA breaks move to the nucleolar periphery for repair [76], and these data suggest that DSBs induced by CX-5461 may be associated with rDNA. Indeed, subsequent analysis of the distribution of DSBs induced by CX-5461 clearly demonstrated their dependence on Top2α and preferential localization to the rDNA promoter/5′ETS regions (Figure 4).

Thus, our results suggest that CX-5461 induces the formation of DNA double-strand breaks localized in the rDNA promoter/5′ETS region by inhibiting Top2α, which is part of the Pol I initiation complex [35]. Therefore, rDNA-localized DSBs cause robust activation of the ATM-dependent DNA damage response [16] without causing extensive DNA damage during the initial stages of drug treatment (non-canonical DDR). This distinguishes CX-5461 from other Top2α poisons and, at least in part, explains the distinct clinical properties of CX-5461 discussed earlier.

In conclusion, we propose that inhibition of rRNA synthesis by CX-5461 results in NSR activation and p53 stabilization. In parallel, CX-5461 induces the formation of double-stranded DNA breaks localized in the rDNA promoter/5′ETS region by inhibiting Pol I-associated Top2α. rDNA-localized DSBs robustly activate the ATM-dependent DNA damage response, leading to p53 phosphorylation and induction of p53 downstream targets. We hypothesize that p53 integrates signals from two pathways (DDR and NSR), causing an amplification of the apoptotic response. In our CX-5461 MOA model, we propose that the location of Top2α double-stranded DNA breaks, rather than the total number of breaks, is critical for efficient CX-5461-mediated Top2α-dependent DDR activation and therefore for CX-5461 anticancer efficacy.

Importantly, the rapid appearance of DSBs following CX-5461 treatment, leading to DDR activation, along with the rapid inhibition of Pol I transcription, triggering NSR, may significantly influence CX-5461 efficacy in vivo. Unlike the sustained drug exposure observed in a large proportion of cellular model experiments, in vivo drug concentrations fluctuate relatively rapidly from zero to high and back to zero before, during and after drug administration. We hypothesize that during this relatively short spike in CX-5461 level, rapid activation of both DDR and NSR occurs, resulting in the rapid and robust activation of apoptosis followed by cell death. Investigating the level and distribution of CX-5461-induced DSBs in cell models (using spiked drug treatment) or animal models could provide valuable insights into a drug’s MOA.

Interestingly, our model and the model proposed by Bossaert and colleagues [23] are not mutually exclusive. Indeed, both models suggest that Top2α-dependent DSBs play an important role in the cytotoxicity of CX-5461. Our model suggests that DSBs are initially formed at the promoter/5′ETS region of active rDNA repeats in a Top2α-dependent manner. This does not exclude a role for CX-5461-dependent stabilization of G4 structures at later stages or at higher concentrations of CX-5461. Furthermore, we have shown that prolonged CX-5461 treatment (Figure 4D) leads to the appearance of DSBs in non-transcribed regions of rDNA (IGS), suggesting the potential extension of DNA damage to other genomic regions.

Our results, alongside the proposed CX-5461 MOA, underscore the pivotal role of targeted DNA damage and transcriptional inhibition of Pol I. This dual functionality of CX-5461 not only defines its therapeutic potential but also anticipates potential side effects. Notably, the recent publication by Koh and colleagues has unveiled the strikingly high mutagenic effect of CX-5461 in cellular models representing both malignant and non-malignant cells [77]. While this study remains confined to in vitro models, it accentuates the paramount importance of assessing the mutagenic burden induced by DNA-damaging drugs. However, it is equally paramount to evaluate the mutagenic burden in vivo. Interestingly, unlike many known mutagens, CX-5461 has no detectable ability to intercalate DNA [78]. With respect to CX-5461, it is imperative that we explore dosing strategies to elicit the acute rDNA-directed apoptotic response while minimizing the potential effects of chronic exposure. Furthermore, to unlock the full clinical potential of targeting Pol I transcription in cancer, we need more selective Pol I inhibitors, and one example of such molecules is the recently developed second-generation Pol I inhibitor PMR-116, which is currently in Phase I clinical trials (ACTRN12620001146987) in Australia.

## Figures and Tables

**Figure 1 biomedicines-12-01514-f001:**
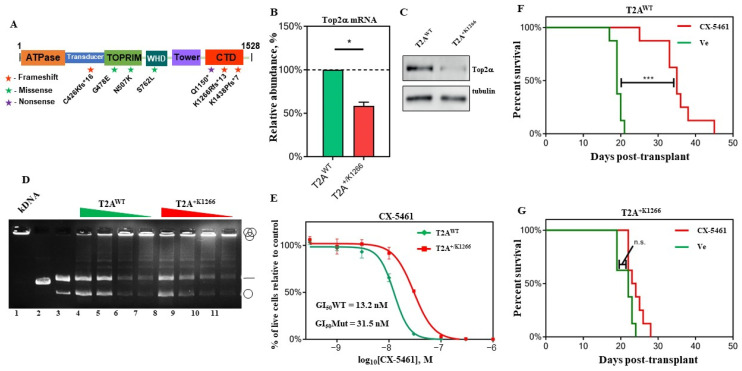
Top2α mediates resistance to CX-5461 in mice. (**A**) Schematic of Top2α mutations. The mouse Top2α gene is shown with the N-terminal ATPase, catalytic TOPRIM, DNA-binding winged-helix domain (WHD), and C-terminal domain (CTD) domains highlighted. Seven different Top2α mutations were identified by whole exome sequencing (WES) of DNA extracted from tumors that had acquired resistance to CX-5461 in vivo. Frameshift, missense and nonsense mutations are denoted by stars, circles, and hexagons, respectively. This figure is adapted from Wendorff et al. [40]. (**B**) Top2α mRNA levels in T2A^WT^ and T2A^+/K1266^ cells. Basal Top2α mRNA abundance in the T2A^WT^ and T2A^+/K1266^ cells was measured by real-time quantitative PCR (qPCR) and normalized to B2M expression. Error bars represent standard deviation (* *p* < 0.05, n = 3). (**C**) Top2α protein expression in T2A^WT^ and T2A^+/K1266^ cells. Representative Western blot of basal Top2α protein. (**D**) The representative image of decatenation assays demonstrating that T2A^WT^ cells have greater cellular Top2 activity than T2A^+/K1266^ cells. The nuclear extracts containing Top2 enzymes are isolated from each group of cells and are serially diluted before incubation with kDNA. In both instances shown, the Top2 activity as measured by the decatenated fraction of each lane is greater in the lower dilutions of the T2A^WT^ cells than the T2A^+/K1266^ cells. (**E**) CX-5461 in vitro dose–response assay. T2A^WT^ (green line) and T2A^+/K1266^ (red line) cells were treated with 0.3 nM–1 μM CX-5461 for 24 h in triplicate, and the live cell number was counted by volumetric FACS of propidium iodide (PI)-negative cells. Results were normalized to the number of live cells after drug vehicle treatment (100%), and the line of best fit was plotted. Concentrations of the drug inhibiting 50% of growth (GI_50_) were calculated using GraphPad Prism 10. (**F**,**G**) The Kaplan–Meier curve of mice transplanted with T2A^WT^ (**F**) and T2A^+/K1266^ (**G**) cells ± CX-5461. Mice were treated every three days with 35 mg/kg of CX-5461. Median survival: T2A^WT^ = 15 days, T2A^+/K1266^ = 1 day (*** *p* < 0.001, n.s. = not significant, n = 8/group).

**Figure 2 biomedicines-12-01514-f002:**
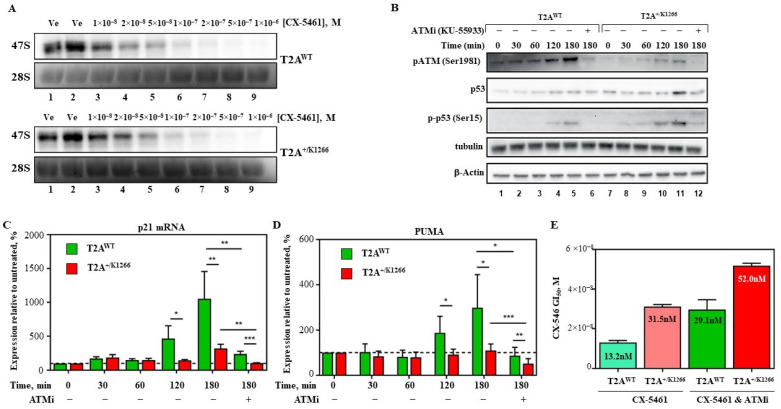
Top2α determines the level of activation of DDR pathways in CX-5461-treated cells but not the ability of CX-5461 to inhibit Pol I transcription. (**A**) T2A^WT^ and T2A^+/K1266^ cells were incubated with radiolabeled orthophosphate for 20 min and then treated with varying concentrations of CX-5461 as indicated. RNA was isolated and analyzed on agarose-formaldehyde gel. Newly synthesized 47 pre-rRNA was detected by autoradiography (a representative experiment is shown; upper panel) and total 28S rRNA was detected by ethidium bromide staining (a representative experiment is shown; bottom panel). (**B**) Representative Western blots showing the downstream signaling response to CX-5461 ± KU-55933 treatment of T2A^WT^ and T2A^+/K1266^ cells. Cells were treated with 30 nM CX-5461 for 30–180 min ± 30 min pre-treatment with ATM inhibitor KU-55933. Western blots were probed for phosphorylated ATM (S1981), phosphorylated p53 (S15), total p53 and loading controls tubulin and β-actin. (**C**,**D**) Quantitative real-time PCR of (**C**) p21 and (**D**) Puma from the experiment outlined in (**B**). Error bars represent standard deviation (* *p* < 0.05, ** *p* < 0.01, *** *p* < 0.001, n = 5). Dotted line represents average relative expression value at time point 0. (**E**) T2A^WT^ and T2A^+/K1266^ cells were treated with varying concentrations of CX-5461 (as in Figure 1E) for 24 h after pre-treatment with ±1 μM KU-55933 for 30 min in triplicate. Cell viability was determined by measuring absorption at 570 nm after incubation with Alamar Blue. Results were normalized to vehicle-treated cells (set 100%), and GI_50_ values were determined and plotted as a bar graph. Error bars represent standard deviation (n = 3).

**Figure 3 biomedicines-12-01514-f003:**
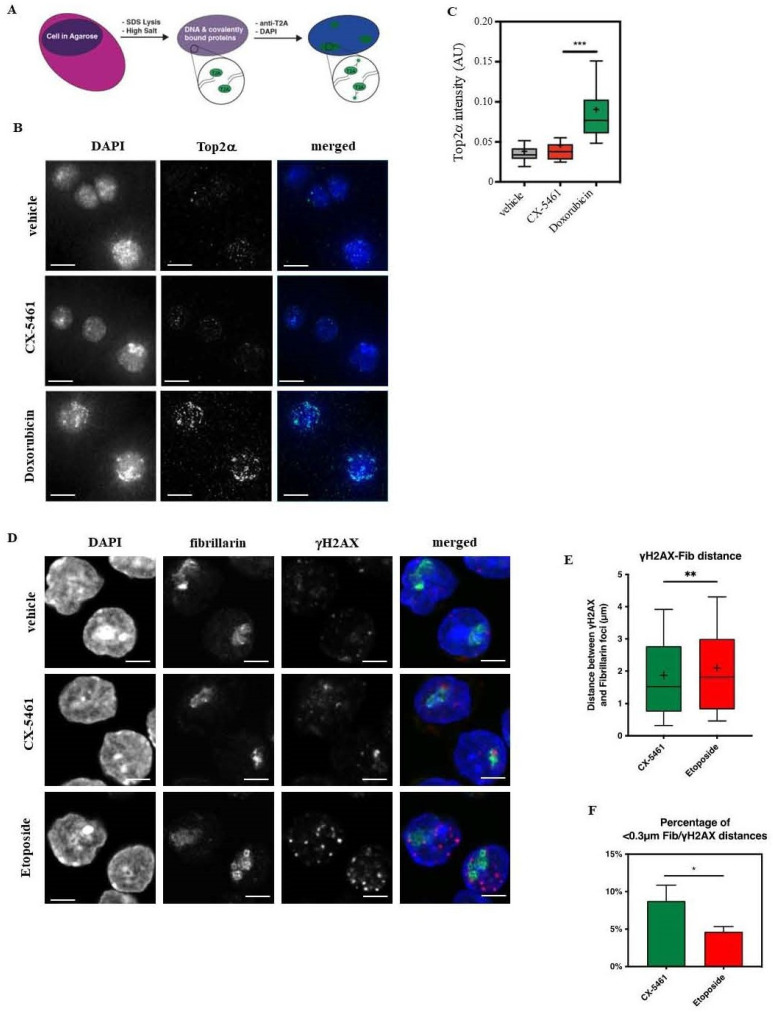
CX-5461 does not induce significant formation of Top2α (Top2α-cc) cleavage complexes, but causes low-level, nucleolar-localized DNA damage. (**A**) Schematic of the TARDIS assay. Cells were suspended in a thin layer of agarose on a microscope slide. After lysis with SDS and incubation with a high salt buffer, only DNA and macromolecules covalently bound to the DNA remain on the slide. Covalently bound proteins, such as active Top2α can be imaged by immunofluorescent labelling. (**B**) Representative images of Eμ-*Myc* T2A^WT^ cells pre-treated with 1 μM MG-132 for 15 min to prevent proteolytic degradation of DNA-bound Top2α, then treated with drug vehicle (top panel), 30-nM CX-5461 (middle panel), or 50-nM doxorubicin (bottom panel) for 2 h. The DNA is stained blue by DAPI, and Top2α bound covalently to DNA is shown in green. Scale bar, 5 μm (**C**) Quantitation of the experiment is outlined in (**B**). While CX-5461 exhibits a similar degree of Top2 inhibition via the decatenation assay (Appendix A), CX-5461 causes significantly fewer Top2α-DNA complexes in cells than doxorubicin (n = 45–47 cells, *** *p* < 0.001). (**D**) Representative images of Eμ-*Myc* T2A^WT^ cells treated with either drug vehicle, 30 nM CX-5461, or 60nM Etoposide for 1 h immunostained for the nuclei (DAPI; blue), the nucleoli (Fibrillarin; green) and γH2AX (red). ). Scale bar, 10.4 μm. (**E**) Quantitation of the distance to the closest fibrillarin spot from each γH2AX foci. Whiskers extend to the 10th and 90th percentiles, and the mean is represented by the cross. CX-5461-treated cells exhibit a shorter median γH2AX-Fib distance compared to vehicle- or etoposide-treated cells (** *p* < 0.01). (**F**) Quantitation of the proportion of sub-0.3 μm γH2AX-Fib distances for both CX-5461 and etoposide-treated samples from three independent experiments. Error bars represent the standard deviation (* *p* < 0.05).

**Figure 4 biomedicines-12-01514-f004:**
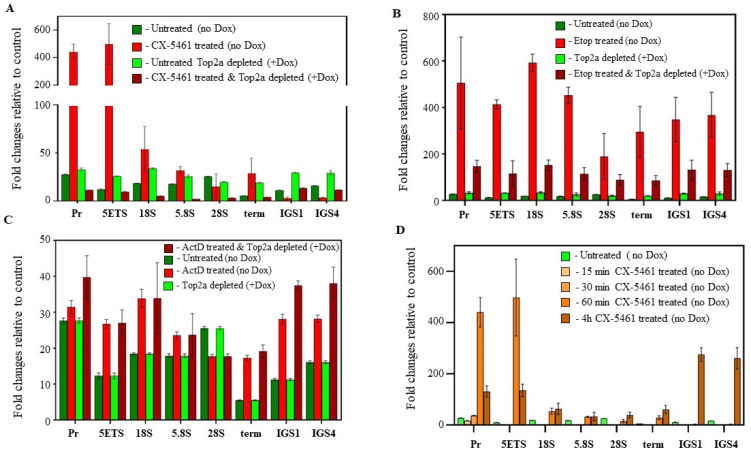
CX-5461 treatment induces targeted, Top2α-mediated DNA damage at the rDNA loci. (**A**) HTETOP cells were grown for 24 h either untreated (−Dox) or in the presence of 1 mg/mL Doxycycline (+Dox) and then treated with ±2 μM CX-5461 for 1 h before the DNA double-strand breaks (DSB) were labelled with BrdUTP using terminal transferase. DNA was extracted and sonicated to 400–600 bps fragments, and the labelled DNA was enriched by anti-BrdUTP antibodies. DNA enrichment was quantitated by qPCR. Signal was normalized to control (IgG) and plotted as fold changes. Error bars represent the standard deviation (n = 3). (**B**) HTETOP cells were grown for 24 h either untreated (−Dox) or in the presence of 1 mg/mL Doxycycline (+Dox) and then treated with ±5 µM etoposide for 2 h. Samples were processed and analyzed as in (**A**). Error bars represent the standard deviation (n = 3). (**C**) HTETOP cells were grown for 24 h either untreated (-Dox) or in the presence of 1 mg/mL Doxycycline (+Dox) and then treated with ±3 nM Actinomyci D (ActD) for 4 h. Samples were processed and analyzed as in (**A**). Error bars represent the standard deviation (n = 3). (**D**) HTETOP cells were grown for 24 h untreated (no Dox) and then treated with 2 μM CX-5461 for different period of time as indicated. Samples were processed and analyzed as in (**A**). Error bars represent the standard deviation (n = 3).

## Data Availability

The raw data supporting the conclusions of this article are not readily available due to technical limitations. Requests to access the data should be directed to Donald Cameron.

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
