# Peer review of "CX-5461 Preferentially Induces Top2α-Dependent DNA Breaks at Ribosomal DNA Loci"

_biomedicines, 2024, doi:10.3390/biomedicines12071514_

Round 1

Reviewer 1 Report

Comments and Suggestions for Authors

The authors have studied in detail the mechanism of action of CX-5461 inhibitor of RNA polymerase I, namely locus-specific induction of double-stranded DNA breaks, stimulating cellular DNA damage response using an impressive set of experimental approaches.

Major points.

The authors investigated in detail the mechanism of action of the CX-5461 inhibitor in Eμ-Myc lymphoma tumor cells and human fibrosarcoma cell line HT1080. Do the authors have data on the effect of CX-5461 on non-cancerous cells? Are non-cancerous cells really less sensitive to CX-5461 than cancer cells?

Could there be other mechanisms of resistance to CX-5461 in addition to reduced Top2α levels in tumors, such as hyperactivation of DNA double-strand break repair pathways? Please discuss these points in the manuscript.

Minor points

Line 278. Please specify, did you use purified chromatin preparation or cell lysates in any way? How were the instrument parameters and processing time to obtain DNA fragments of the specified size?

Reviewer 2 Report

Comments and Suggestions for Authors

In this study the Authors identified Top2α as a key factor influencing response to CX-5461 in a mouse model of Eμ-Myc B lymphoma, with sensitivity to CX-5461 dependent on the expression and activity of cellular Top2α. Notably, unlike typical Top2α poisons, CX-5461 induces Top2α-dependent DNA damage that is primarily concentrated at the ribosomal DNA (rDNA) promoter region. This observation underscores CX-5461 as a DNA-damaging agent with specific action at particular genomic loci

This study is well written and perfectly done.  There are only several minor points.

Abstract: 1. Please, clarify how specific the therapeutic agent CX-5461 is to cancer cells. Does it also damage normal cells? In other words, please explain why inhibiting the ribosome biogenesis process specifically targets cancer cells.

2. Could you please specify in a few words what is special about the "non-classical response to DNA damage caused by CX-5461"? Please briefly describe the methods used. The result "Sensitivity to CX-5461 depends on the expression/activity of cellular Top2α" is very important. Please explain how it depends and whether this activity differs between cancerous and normal cells.

3. Please, fish out small misprints from the text (For example, the caption “3.1.2. Top2α mediate resistance to CX-5461…” line 348   ).

Reviewer 3 Report

Comments and Suggestions for Authors

In this work, the authors revealed the relationship between Topoisomerase 2α (Top2α) and the RNA Polymerase I (Pol I) transcription inhibitor CX-5461. The authors identified the Top2α expression/activity-dependent sensitivity to CX461 in murine Eμ-Myc B lymphoma model and demonstrated the ribosomal DNA (rDNA) promoter region is the target loci of CX-5461-induced Top2α-dependent DNA damage. This work contributed new insights into the mechanism of action of CX-5461 and is of great importance for the novel anti-cancer drug development.

Below are some minor suggestions.

Page 1, line 33: I would recommend changing “Transcription of ribosomal RNA (rRNA) by RNA polymerase I (pol I)…” to “Transcription of ribosomal DNA (rDNA) by RNA polymerase I (pol I)…” to avoid confusion.

Figure 1: Please list the concentrations of CX-5461 employed for the Kaplan-Meier curves in the legends to Fig 1F&1G.

Overall, this manuscript is well organized. All the conclusions have reliable experimental support. The content also contains comprehensive and insightful discussions that enhance the significance of the results and provide new insights into the development of Pol I inhibitors with great clinical potential.

An open question: Does this mean that the combination of Top2α and RNA Pol I inhibition will be a more effective cancer treatment strategy? And is the correlation between Top2α and RNA Pol I cell line dependent? How applicable is it?

Round 2

Reviewer 1 Report

Comments and Suggestions for Authors

The authors have improved the manuscript in accordance with the comments. I have no further significant comments and suggest that the manuscript be accepted for publication in the journal.